# The association of micro and macro worries with psychological distress in people living with chronic kidney disease during the COVID-19 pandemic

Ella C. Ford[1,2¤], Gurneet K. Sohansoha[1,2], Naeema A. Patel[1,2], Roseanne E. Billany[2,3], Thomas J. Wilkinson[1,2], Courtney J. Lightfoot[1,2], Alice C. Smith[1,2]*

1 Leicester Kidney Lifestyle Team, Department of Population Health Sciences, University of Leicester, Leicester, United Kingdom, 2 Leicester NIHR Biomedical Research Centre, Leicester, United Kingdom, 3 Department of Cardiovascular Sciences, University of Leicester, Leicester, United Kingdom

¤ Current address: Department of Psychology, School of Humanities and Social Sciences, Leeds Beckett University, Leeds, United Kingdom
* aa50@le.ac.uk

**Data Availability Statement:** All relevant data are within the manuscript and its Supporting Information files.

## Abstract

### Background

Psychological distress can be exacerbated by micro (personal) and macro (societal) worries, especially during challenging times. Exploration of this relationship in people with chronic kidney disease is limited.

### Objectives

(1) To identify the types and levels of worries concerning people with chronic kidney disease in the context of the COVID-19 pandemic; (2) to explore the association of worries with psychological distress including depression, stress, anxiety, and health anxiety.

### Design and participants

A cross-sectional online survey collected data at two time points (Autumn 2020, n = 528; Spring 2021, n = 241). Participants included kidney transplant recipients and people with non-dialysis dependent chronic kidney disease.

### Measurements

The survey included questions about worry taken from the World Health Organisation COVID-19 Survey, the Depression, Anxiety and Stress Scale, and the Short Health Anxiety Index. Data were analysed using descriptive statistics and multiple regression.

### Results

Worries about loved ones' health, the healthcare system becoming overloaded, losing a loved one, economic recession, and physical health were the highest rated concerns. Worrying about mental health was associated with higher depression, stress, anxiety, and health

**Funding:** Stoneygate Trust Kidney Lifestyle Research Programme 2018-24 (ACS) https://www.stoneygatetrust.org/ The funder played no role in the study design, data collection/analysis, decision to publish or preparation of the manuscript.

**Competing interests:** The authors have declared that no competing interests exist.

anxiety. Worrying about physical health was associated with anxiety and health anxiety. Worrying about losing a loved one was associated with health anxiety, and worrying about not being able to pay bills was associated with stress.

## Conclusions

People with kidney disease reported micro and macro worries associated with psychological distress during the COVID-19 pandemic. This study highlights factors that should be considered to improve the mental health and well-being of people with kidney disease.

## Introduction

Chronic kidney disease (CKD) is a common progressive long-term condition (LTC) defined by reduced kidney function or structural abnormality persisting for at least 3 months. A minority of people with CKD progress to end stage kidney failure requiring renal replacement therapy such as dialysis or transplantation to maintain life. However, the majority live with stable or slowly declining kidney function associated with high co-morbidity and symptom burden, which often includes impaired metabolic and immune function. Although kidney transplantation can provide profound improvements in kidney function, general health and quality of life (QoL), lifelong immunosuppression is required to prevent rejection, with consequent increased susceptibility to infection. For these reasons, people with CKD at all stages, and especially kidney transplant recipients, were deemed to be clinically extremely vulnerable during the COVID-19 pandemic, and advised to take particular precautions to avoid infection.

Psychological distress, such as elevated levels of anxiety, depression, stress, and health anxiety, is prevalent in people living with CKD [1]. High incidence of anxiety and depressive symptoms are observed across the disease trajectory [2], with higher rates of depression observed compared to other LTCs [3]. These elevated levels of distress are associated with poor QoL, faster decline of kidney function, and higher risk of hospitalisation and mortality [1, 2, 4, 5]. Challenging and stressful periods, such as the COVID-19 pandemic, can worsen psychological distress [6, 7], which negatively impacted the mental health and QoL of people with CKD [8–12]. Worry is a common reaction to threat [13], and is associated with psychological distress [14, 15]. Understanding the worries of people with CKD during the pandemic and how these relate to psychological distress may help to better comprehend how to minimise the impact of challenging circumstances on mental health and well-being.

Worry is often defined as apprehensive expectations and repetitive negative thoughts about risks, threats, and uncertainties regarding the future [16, 17]. Worry comprises of two primary dimensions: process and content [18]. Content can be categorised into micro (related to the self and in-group) and macro (related to society) worries [19, 20]. Whilst worrying can represent the initial stages of problem-solving [16, 19], when elevated and non-constructive, it can become maladaptive to well-being [17]. Substantial perceived risks, including those experienced during the COVID-19 pandemic, can increase worry [13]. This was exacerbated for people with CKD who reported heightened risk perception [21], likely due to their increased risk of serious illness from contracting COVID-19 [22], and the substantial changes to their healthcare and daily routines [23, 24].

COVID-19-related worries have been associated with illness-related distress and depressive symptoms in people with LTCs [25], poor mental and health-related QoL in dialysis recipients [26], and perceived stress in people with CKD [27]. Worry is typically considered as a

collective whole; however, worry is not unitary. A multidimensional approach concerning the content of worries may elicit a more comprehensive understanding, particularly as different worries can have different effects on mental health [18] and behaviours in the context of stressful events [28]. For example, in kidney transplant recipients (KTRs) and their donors, worries about their household's physical health and their own mental health in particular were associated with higher psychological distress [29].

There is a dearth of studies examining the relationship between psychological distress and worry in the CKD population, particularly regarding those with non-dialysis CKD (NDCKD) who represent a substantial proportion of this population [30]. Examining this relationship may provide insight into the most prominent and important issues for people with CKD, enabling healthcare professionals (HCPs) to consider these factors in a holistic treatment model. The impacts of the pandemic are yet to be determined [31, 32], but those with CKD are likely to continue to be affected [33], particularly as many worry-inducing issues which arose during the pandemic are still ongoing [32] including the threat of economic recession and deterioration of healthcare-systems [34]. Levels of worry were observed to remain stable throughout the pandemic [31, 32]; consequently, worry has not necessarily declined with the relaxing of restrictions and vaccination efforts.

Therefore, the present study aimed to identify the types and levels of worries concerning people living with CKD, and explore their associations with levels of psychological distress, including depression, anxiety, stress and health anxiety, during the COVID-19 pandemic.

## Materials & methods

### Design and setting

The study used a cross-sectional survey design. Data were collected via Jisc Online Surveys between August 2020 and June 2021 as part of the larger multicentre study DIMENSION-KD (ISRCTN84422148) which was adapted in 2020 in response to the World Health Organisation (WHO) declaring COVID-19 a global pandemic [35]. The study was approved by the Leicester Research Ethics Committee (24/05/2018, reference: 18/EM/0117) and prospectively registered as ISRCTN84422148 in June 2018. The study was conducted in accordance with the Declaration of Helsinki and local and national ethical guidelines. All participants voluntarily participated in the study and provided informed online consent.

Data were collected in England, United Kingdom, at two time points: time point 1 (T1) between 1st August 2020 and 30th November 2020 when England was under tiered or full lockdown restrictions; time point 2 (T2) between 1st May and 30th June 2021, during partial lockdown restrictions and COVID-19 vaccination roll out (S1 Text). Participants were identified by their local healthcare teams and invited by letter by local research staff. The letter contained a link to an online survey consisting of two parts. The first part included questions relating to demographic factors and COVID-19, while the optional second part included validated questionnaires regarding psychological distress. Participants who completed any part of the T1 survey were sent an email invitation at T2 containing a link to the T2 survey, consisting of both a reduced subset of bespoke COVID-19 questions and the validated questionnaires.

### Participants

Participants were recruited from 11 hospital sites across England and were under the management of a consultant nephrologist. The inclusion criteria were: (1) aged ≥ 18 years; (2) diagnosis of an established kidney condition (stages 1–5) not requiring dialysis but including KTRs; (3) ability to provide informed consent and adhere to study protocol. Power calculations to inform a sample size target were not performed as the study was created to inform a rapid

evidence synthesis during the COVID-19 pandemic; the aim was to recruit the maximum number of participants within the recruitment timeframes.

## Measurements

**Sociodemographic.** Self-reported sociodemographic variables were collected at T1. Age, sex, ethnicity, education, pre-COVID-19 employment status, type of kidney condition (NDCKD or KTR), comorbidities, and socioeconomic status (SES) (measured by index of multiple deprivation decile (IMDD) identified by postcode) were collected. The estimated glomerular filtration rate (eGFR) was extracted from medical records.

**Worries.** Responses to 13 individual statements revised from the WHO COVID-19 survey guidance [36] (adapted from McCarthy-Larzelere et al [18]) (S2 Text) were used to assess the level of worries about concepts such as health, restrictions on movement, personal economy, and their country's economy [37]. Each item was measured on a 7-point Likert scale from 1 ('do not worry at all') to 7 ('worry a lot').

**Depression, anxiety and stress.** Depression, anxiety, and stress were assessed using the self-report Depression, Anxiety and Stress Scale (DASS-21) which measures negative emotional states across the three dimensions [38]. Each dimension contains 7 items measured on a Likert scale of 0 ('never') to 3 ('almost always'). The depression dimension measures symptoms regarding hopelessness, lack of self-esteem, and motivation. The anxiety dimension concerns symptoms around autonomic arousal and situational and subjective feelings of fear and anxiety. The stress dimension considers symptoms relating to persistent arousal, tension, and irritability [39]. The maximum sum of scores for each dimension is 21, with higher scores indicating higher levels of depression, anxiety, and stress. The DASS has good reliability and validity in community and clinical populations [40] and has been used in research conducted during the COVID-19 pandemic [41]. The Cronbach's alphas were 0.92 (Depression) 0.83 (Anxiety) 0.90 (Stress) at T1, and 0.92 (Depression) 0.79 (Anxiety) 0.90 (Stress) at T2.

**Health anxiety.** Health anxiety was assessed using the Short Health Anxiety Index (SHAI) [42]. The SHAI consists of 14 items assessing health anxiety characteristics such as concern about health, awareness of bodily sensations, and feared consequences of illness [43]. Each item is weighted on a scale from 0 to 3. The overall score was calculated by summing the items with a maximum score of 42. Higher scores are indicative of higher levels of health anxiety. The SHAI has been shown to have good reliability and validity in clinical populations [42]. The Cronbach's alpha was 0.89 at T1 and 0.90 at T2.

**Statistical analysis.** Data were analysed using IBM SPSS Statistics (Version 28). Descriptive statistics were calculated giving absolute and relative frequencies regarding qualitative variables, and the mean, standard deviation (SD), and range for quantitative variables. Data are presented as mean and SD unless otherwise stated. Statistics are presented for the whole data set, as well as for those included in the following analyses at T1 and T2.

Multivariate multiple regression analyses were used to assess the association of worries, the independent variables as measured by the WHO [36] questionnaire, with the dependent variables of depression, anxiety, and stress, as measured by the DASS-21, and health anxiety, as measured by the SHAI (i.e., does worry explain the variance in depression, anxiety, stress, and health anxiety). Participants who had complete data regarding these variables were included at each time point. Worry about becoming unemployed was not included in the analysis as many participants were not employed, descriptive data regarding the worries of employed participants can be found in S1 Table.

All models were adjusted for age, gender, type of kidney condition, and SES (IMDD) based on the importance these factors have had on measures of worry and psychological distress

[7, 37]. All worry variables were entered into Model 1, and covariates were additionally entered into Model 2. Multivariate regressions were conducted individually for each dependent variable at both T1 and T2 separately; worry variables measured at T1 were entered with the dependent variables measured at T1, and similarly at T2. Results from the adjusted model (Model 2) from each regression are reported, results from the unadjusted model (Model 1) can be found in S2 Table. Inspection of the ZPRED vs ZRESID plots suggested a degree of heteroscedasticity, therefore all analyses were conducted were bootstrapped using 1000 samples and the data reported was based on 95% bias-corrected and accelerated. Data are presented as standardised coefficient betas and significance values. Significance was recognized as p < .050.

## Results

### Participant characteristics

528 participants (55.5% KTR; 45.5% NDCKD; age: 60.5±12.8 years (18–90); IMDD: 6.6±2.5 (1–10); 56.2% male; 92.8% White British) completed the survey at T1 and n = 241 at T2. Participants who completed the variables of interest were included in the regression analyses: n = 245 at T1, and n = 224 at T2 (details of completion rates are shown in Fig 1). Participant characteristics are displayed in Table 1. A large number of T1 participants were not included as they did not complete the optional second part of the survey (containing the DASS-21 and

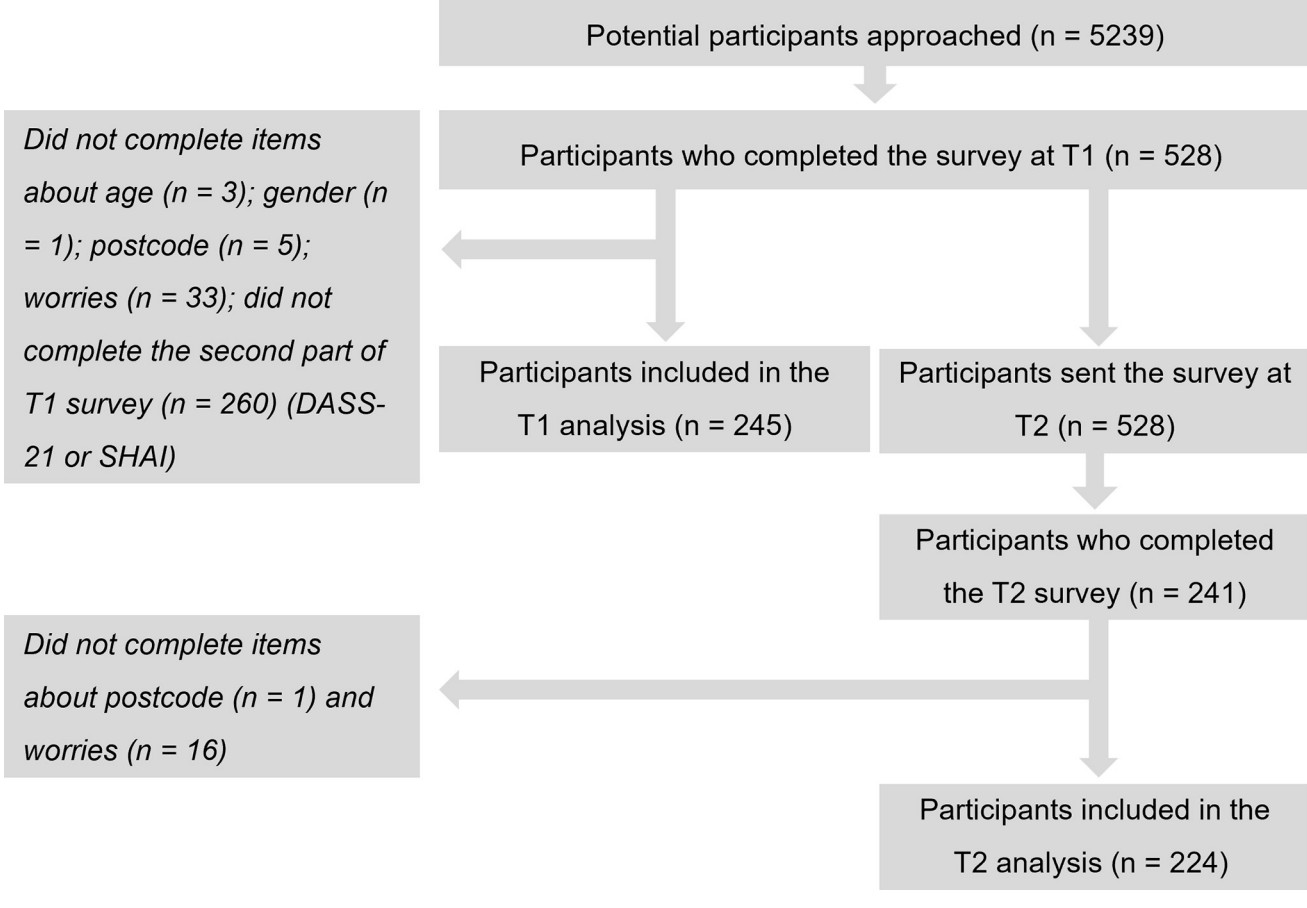

**Fig 1. Modified CONSORT diagram of participants.** Reasons for decline to complete the survey and elements of the survey were not collected. Abbreviations: DASS-21, Depression, Anxiety and Stress Scale– 21; SHAI, Short Health Anxiety Index; T1, timepoint 1; T2, timepoint 2.

**Table 1. Demographic characteristics at T1 of all participants and those included in the multiple regression analyses at T1 and T2.**

| Demographic | Total (N = 528) | | T1 (N = 245) | | T2 (N = 224) | |
|---|---|---|---|---|---|---|
| **Age** | 60.5 | ± 12.8 | 61.2 | ± 13.3 | 63.4 | ± 11.4 |
| **IMDD** | 6.6 | ± 2.5 | 6.8 | ± 2.4 | 6.9 | ± 2.4 |
| **Type of Kidney Problem** | | | | | | |
| NDCKD | 235 | (44.5%) | 112 | (45.7%) | 113 | (50.5%) |
| eGFR, ml/min/1.73m2 | 35.9 | ± 20.8 | 38.2 | ± 21.3 | 36.6 | ± 19.0 |
| KTR | 293 | (55.5%) | 133 | (54.3%) | 111 | (49.6%) |
| **Gender** | | | | | | |
| Female | 231 | (43.8%) | 108 | (44.1%) | 94 | (42.0%) |
| Male | 296 | (56.2%) | 137 | (55.9%) | 130 | (58.0%) |
| **Ethnicity** | | | | | | |
| White British | 490 | (92.8%) | 228 | (93.1%) | 212 | (94.6%) |
| Other White | 13 | (2.5%) | 8 | (3.3%) | 5 | (2.2%) |
| South Asian | 12 | (2.3%) | 8 | (3.3%) | 3 | (1.3%) |
| Other Ethnicity | 13 | (2.5%) | 1 | (0.4%) | 4 | (1.8%) |
| **Education** | | | | | | |
| None | 21 | (4.0%) | 5 | (2.0%) | 10 | (4.5%) |
| High school | 131 | (24.9%) | 52 | (21.2%) | 44 | (19.6%) |
| College | 153 | (29.1%) | 74 | (30.2%) | 66 | (29.5%) |
| Trade qualification | 57 | (10.8%) | 30 | (12.3%) | 23 | (10.3%) |
| University | 164 | (31.2%) | 84 | (34.3%) | 80 | (35.7%) |
| **Employment Status** | | | | | | |
| Employed | 190 | (36.0%) | 81 | (33.1%) | 68 | (30.4%) |
| Self-employed | 55 | (10.4%) | 24 | (9.8%) | 27 | (12.1%) |
| Retired | 249 | (47.2%) | 125 | (51.0%) | 120 | (53.6%) |
| Unemployed | 14 | (2.7%) | 6 | (2.5%) | 3 | (1.3%) |
| Carer / Homemaker | 11 | (2.1%) | 4 | (1.6%) | 3 | (1.3%) |
| Student | 6 | (1.1%) | 4 | (1.6%) | 2 | (0.9%) |
| Other | 13 | (2.5%) | 6 | (2.5%) | 5 | (2.2%) |
| **Comorbidities** | | | | | | |
| Hypertension | 407 | (77.1%) | 193 | (78.8%) | 181 | (80.8%) |
| Diabetes type II | 98 | (18.6%) | 43 | (17.6%) | 43 | (19.2%) |
| Mental health issues | 105 | (19.9%) | 45 | (18.4%) | 28 | (12.5%) |

IMDD, index of multiple deprivation decile; eGFR, estimated glomerular filtration rate; NDCKD, non-dialysis chronic kidney disease; KTR, kidney transplant recipient; T1, time point 1; T2, time point 2. Data are presented as mean ± SD, or $n$ (%)

SHAI), however, these participants did not greatly differ demographically from participants who were included (S3 Table).

**Worries.** The mean participant ratings for each WHO survey 'worry' item are displayed in Table 2.

The top three highest-rated worry items for total participants were worries about 'loved one's health', 'healthcare system becoming overloaded', and 'losing a loved one'. Worry about 'economic recession' and 'physical health' were the next highly rated. All mean ratings for these top five worries rounded to the midpoint of ~4-5/7 at both time points.

## Levels of psychological distress

Table 3 shows the mean DASS-21 components and SHAI scores of those included in the multiple regression analyses at T1 and T2.

**Table 2. Mean ratings of worry items for all participants at T1, and those included in the multiple regression analyses at T1 and T2.**

| Worry | | Total (*N* = 528) | | T1 (*N* = 245) | | T2 (*N* = 224) | |
|---|---|---|---|---|---|---|---|
| | n | mean | ± SD | mean | ± SD | mean | ± SD |
| Losing a loved one | 521 | 4.8 | 1.9 | 4.8 | 1.9 | 4.4 | 1.8 |
| Healthcare system becoming overloaded | 520 | 4.9 | 1.7 | 4.8 | 1.7 | 4.6 | 1.6 |
| Mental health | 520 | 3.3 | 2.0 | 3.0 | 2.0 | 2.9 | 1.8 |
| Physical health | 521 | 4.4 | 1.9 | 4.3 | 2.0 | 4.0 | 1.8 |
| Loved one's health | 523 | 5.2 | 1.7 | 5.2 | 1.8 | 4.8 | 1.6 |
| Restriction of movement | 520 | 3.6 | 2.0 | 3.4 | 2.0 | 2.9 | 1.8 |
| Losing holiday opportunities | 519 | 2.8 | 2.0 | 2.8 | 2.0 | 2.2 | 1.6 |
| Economic recession | 521 | 4.4 | 1.8 | 4.5 | 1.8 | 3.6 | 1.7 |
| Restricted access to essential supplies | 519 | 4.1 | 1.9 | 3.9 | 1.9 | 3.4 | 1.7 |
| Not being able to pay bills | 514 | 2.8 | 2.1 | 2.6 | 2.0 | 2.3 | 1.7 |
| Not being able to visit dependents | 510 | 3.3 | 2.0 | 3.1 | 2.0 | 2.9 | 1.9 |
| Defending not socially participating | 518 | 2.5 | 1.9 | 2.3 | 1.8 | 2.0 | 1.6 |

T1, time point 1; T2, time point 2.

## Associations between worries and demographic factors with measures of psychological distress

The association of worries with depression, anxiety, stress, and health anxiety are reported in Table 4 (T1 analysis), and Table 5 (T2 analysis). At T1, increased worry about mental health was associated with higher depression, stress, anxiety, and health anxiety. Increased worry about losing a loved one was associated with higher health anxiety. Worrying about physical health was associated with higher anxiety and higher health anxiety. Worrying about not being able to pay bills was associated with higher stress. Three of these associations remained significant at T2: worrying about mental health was associated with higher depression and stress; worrying about physical health was associated with higher health anxiety.

## Discussion

The present study examined the different types and levels of worries in people living with CKD during the COVID-19 pandemic, and the associations of these worries with psychological distress including depression, anxiety, stress, and health anxiety. The highest-rated worries were micro worries about the health and lives of their loved ones, which is consistent with adults in the general population [23], and with people across stages of CKD [10, 44]. The next highest-rated were macro worries about healthcare system overload and economic recession, consistent with research into other LTCs [25]. Worries about mental health, physical health,

**Table 3. Mean rating of DASS-21 component scores and SHAI scores at T1 and T2.**

| | T1 (*N* = 245) | | | | T2 (*N* = 224) | | | |
|---|---|---|---|---|---|---|---|---|
| | Depression | Anxiety | Stress | SHAI | Depression | Anxiety | Stress | SHAI |
| N | 240 | 241 | 244 | 219 | 221 | 221 | 216 | 208 |
| MEAN | 4.2 | 2.5 | 4.5 | 12.3 | 4.0 | 2.7 | 4.2 | 11.0 |
| ± SD | 4.4 | 3.4 | 4.2 | 6.7 | 4.5 | 3.3 | 4.3 | 7.0 |
| RANGE | 0–21 | 0–21 | 0–21 | 1–35 | 0–21 | 0–19 | 0–21 | 0–33 |

DASS-21, Depression, Anxiety and Stress Scale– 21; SHAI, Short Health Anxiety Inventory; T1, time point 1; T2, time point 2.

**Table 4. Association between worries at T1 and depression, anxiety, stress and health anxiety at T1 (N = 245).**

| | Depression | | | Anxiety | | | Stress | | | SHAI | | |
|---|---|---|---|---|---|---|---|---|---|---|---|---|
| **Worry** | (N = 240) | | | (N = 241) | | | (N = 244) | | | (N = 219) | | |
| | β | CI | p | β | CI | p | β | CI | p | β | CI | p |
| Losing a loved one | 0.05 | (-0.25, 0.51) | 0.550 | -0.04 | (-0.34, 0.19) | 0.621 | 0.02 | (-0.35, 0.38) | 0.841 | 0.22 | (0.23, 1.32) | **0.004*** |
| Healthcare system becoming overloaded | 0.07 | (-0.20, 0.47) | 0.308 | 0.11 | (-0.07, 0.44) | 0.163 | 0.12 | (-0.03, 0.61) | 0.083 | 0.06 | (-0.28, 0.77) | 0.332 |
| Mental health | 0.45 | (0.57, 1.36) | **< .001*** | 0.31 | (0.19, 0.87) | **0.008*** | 0.46 | (0.56, 1.29) | **< .001*** | 0.23 | (0.15, 1.32) | **0.007*** |
| Physical health | 0.04 | (-0.25, 0.46) | 0.640 | 0.18 | (0.04, 0.57) | **0.013*** | 0.06 | (-0.18, 0.45) | 0.464 | 0.37 | (0.77, 1.76) | **< .001*** |
| Loved one's health | -0.03 | (-0.52, 0.36) | 0.783 | -0.02 | (-0.35, 0.28) | 0.867 | 0.01 | (-0.42, 0.44) | 0.936 | -0.09 | (-1.01, 0.31) | 0.304 |
| Restriction of movement | 0.09 | (-0.21, 0.62) | 0.327 | -0.13 | (-0.55, 0.10) | 0.196 | -0.06 | (-0.45, 0.21) | 0.477 | -0.03 | (-0.70, 0.51) | 0.717 |
| Losing holiday opportunities | -0.02 | (-0.34, 0.22) | 0.759 | -0.01 | (-0.22, 0.18) | 0.916 | -0.03 | (-0.32, 0.18) | 0.661 | -0.05 | (-0.61, 0.29) | 0.498 |
| Economic recession | -0.13 | (-0.65, 0.03) | 0.058 | -0.08 | (-0.49, 0.24) | 0.353 | -0.09 | (-0.52, 0.18) | 0.224 | -0.11 | (-1.01, 0.14) | 0.103 |
| Restricted access to essential supplies | 0.04 | (-0.23, 0.42) | 0.573 | -0.01 | (-0.30, 0.23) | 0.894 | 0.02 | (-0.24, 0.30) | 0.794 | 0.01 | (-0.41, 0.50) | 0.846 |
| Not being able to pay bills | 0.08 | (-0.16, 0.48) | 0.367 | 0.15 | (-0.05, 0.52) | 0.121 | 0.16 | (0.03, 0.63) | **0.038*** | 0.06 | (-0.32, 0.66) | 0.509 |
| Not being able to visit dependents | -0.05 | (-0.38, 0.12) | 0.406 | 0.07 | (-0.12, 0.32) | 0.315 | 0.00 | (-0.20, 0.22) | 0.938 | 0.00 | (-0.43, 0.47) | 0.962 |
| Defending not socially participating | 0.01 | (-0.36, 0.39) | 0.915 | -0.01 | (-0.26, 0.24) | 0.902 | 0.06 | (-0.17, 0.41) | 0.379 | 0.06 | (-0.36, 0.83) | 0.433 |

β, standardised coefficient; CI, confidence interval; T1, time point 1.

Adjusted for sex, age, index of multiple deprivation decile, and type of kidney disease. Confidence intervals are 95% bias corrected and accelerated. Confidence intervals and p values are based on 1000 bootstrapped samples. Significance (*) p<0.05

ability to pay bills, and losing loved ones were associated with psychological distress. This understanding can aid HCPs in holistic care assessment to better support these individuals.

The most prominent worries were rated on average 4-5/7, suggesting a moderately high level of worry. Like Kim et al [25], we found that worry about economic recession (macroeconomic) was one of the highest rated concerns, despite worries about personal finances (microeconomic) being some of the lowest rated. This could be due to the sample's high average SES.

**Table 5. Association between worries at T2 and depression, anxiety, stress and health anxiety at T2 (N = 224).**

| | Depression | | | Anxiety | | | Stress | | | SHAI | | |
|---|---|---|---|---|---|---|---|---|---|---|---|---|
| **Worry** | (N = 221) | | | (N = 221) | | | (N = 216) | | | (N = 208) | | |
| | β | CI | p | β | CI | p | β | CI | p | β | CI | p |
| Losing a loved one | 0.11 | (-0.34, 0.80) | 0.342 | 0.11 | (-0.19, 0.62) | 0.334 | 0.02 | (-0.48, 0.58) | 0.829 | 0.08 | (-0.67, 1.20) | 0.514 |
| Healthcare system becoming overloaded | 0.04 | (-0.44, 0.60) | 0.686 | 0.07 | (-0.21, 0.51) | 0.383 | 0.09 | (-0.23, 0.73) | 0.255 | 0.04 | (-0.60, 1.03) | 0.634 |
| Mental health | 0.37 | (0.39, 1.51) | **< .001*** | 0.21 | (0.00, 0.76) | 0.052 | 0.31 | (0.25, 1.28) | **0.003*** | 0.17 | (-0.13, 1.48) | 0.092 |
| Physical health | 0.08 | (-0.26, 0.60) | 0.371 | 0.12 | (-0.08, 0.48) | 0.220 | -0.01 | (-0.46, 0.38) | 0.955 | 0.24 | (-0.04, 1.73) | **0.036*** |
| Loved one's health | -0.09 | (-0.87, 0.48) | 0.434 | 0.05 | (-0.34, 0.55) | 0.680 | 0.12 | (-0.26, 0.93) | 0.282 | 0.01 | (-0.96, 1.17) | 0.934 |
| Restriction of movement | -0.02 | (-0.52, 0.44) | 0.809 | -0.17 | (-0.67, 0.03) | 0.063 | -0.05 | (-0.57, 0.35) | 0.572 | -0.07 | (-0.93, 0.39) | 0.450 |
| Losing holiday opportunities | -0.05 | (-0.57, 0.32) | 0.511 | 0.04 | (-0.24, 0.40) | 0.566 | -0.02 | (-0.52, 0.44) | 0.791 | 0.01 | (-0.69, 0.84) | 0.899 |
| Economic recession | 0.07 | (-0.28, 0.56) | 0.468 | 0.07 | (-0.17, 0.43) | 0.440 | 0.03 | (-0.37, 0.47) | 0.748 | 0.01 | (-0.67, 0.68) | 0.930 |
| Restricted access to essential supplies | -0.06 | (-0.62, 0.36) | 0.567 | -0.05 | (-0.45, 0.26) | 0.657 | 0.03 | (-0.40, 0.52) | 0.818 | -0.04 | (-0.98, 0.71) | 0.682 |
| Not being able to pay bills | 0.03 | (-0.45, 0.64) | 0.778 | -0.03 | (-0.41, 0.32) | 0.739 | 0.02 | (-0.51, 0.68) | 0.846 | -0.06 | (-0.86, 0.43) | 0.540 |
| Not being able to visit dependents | -0.02 | (-0.40, 0.32) | 0.807 | 0.04 | (-0.23, 0.36) | 0.679 | 0.07 | (-0.21, 0.51) | 0.425 | -0.05 | (-0.72, 0.39) | 0.473 |
| Defending not socially participating | 0.06 | (-0.38, 0.62) | 0.522 | 0.17 | (-0.06, 0.79) | 0.111 | 0.02 | (-0.47, 0.48) | 0.885 | 0.12 | (-0.23, 1.19) | 0.176 |

β, standardised coefficient; CI, confidence interval; T2, time point 2.

Adjusted for sex, age, index of multiple deprivation decile, and type of kidney disease. Confidence intervals are 95% bias corrected and accelerated. Confidence intervals and p values are based on 1000 bootstrapped samples. Significance (*) p<0.05

This is an important insight as few studies have considered macroeconomic worries and their effects, instead focusing on microeconomic worries such as medical costs [29, 44]. As people in the United Kingdom are increasingly reporting financial strain due to increases in private healthcare use and travelling distance for appointments [45, 46], these worries should be considered in future research.

These perceived increases in personal medical costs are arguably partly due to difficulties in accessing efficient and timely healthcare [45, 46], which is reflected in the prominence of worry about healthcare system overload in the sample. At the time of the study, there had been a considerable decrease in CKD care provisions and quality due to COVID-19 [47, 48]. Many associated problems are ongoing, such as staff shortages and lengthening waiting lists, increasing pressures on already overwhelmed health services [49, 50]. For individuals with CKD who rely on life-long, frequent healthcare contact [30, 33], worries about the capacity of services can lead to uncertainty in accessing care and can result in increased emotional distress and burden [46]. These worries could also negatively affect relationships with HCPs and the utilisation of services [47], and contribute to negative expectations of healthcare interactions [45, 51]. HCPs and policymakers should consider worries regarding the healthcare system and ways to reduce these to mitigate negative consequences.

We found worries about mental health, physical health, ability to pay bills, and losing loved ones to be associated with psychological distress. Levels of psychological distress are considerably low among our study population in comparison to other studies conducted during the COVID-19 pandemic [10, 52]. This could be due to the sample consisting of mostly White British participants of high SES, as people with CKD of non-White ethnicities and lower SES have reported higher rates of mental health problems [53]. There were differences in the significance of associations between worries and psychological distress observed at T1 compared to T2. Various factors could have contributed to this. For example, there were less participants who self-reported a mental health condition at T2 compared to at T1 (a 32% relative decrease), and around half of participants at T1 did not complete T2 and vice versa. This difference could also be due to changes in restrictions and understanding of COVID-19 between time points. The worries associated with distress at T1 could also have had more relevance to distress at the beginning of the stressful event when levels of uncertainty are elevated.

Worry about mental health was associated with all measures of psychological distress at T1, and depression and stress at T2, this link is consistent with previous research with KTRs [29]. Understandably, those worrying about their mental health may be more likely to have symptoms of these mental health problems and vice versa. It is imperative that HCPs proactively identify those with mental health related worries to appropriately support them by mitigating the effects of further worry and distress on their well-being and CKD management [1, 2, 4, 5]. Worrying about physical health was associated with health anxiety at both time points and with anxiety at T1; this is expected as health anxiety is often characterised by excessive worry about disease [54]. Lower levels of this worry can motivate people to improve their health [55], however excessive worry and health anxiety can negatively impact QoL and limit daily functioning [56, 57]. This is especially relevant in a post-pandemic context where people may consequentially be arguably overcautious, limiting their participation in interactions involving potential infection risk [18, 58] such as accessing healthcare settings. Conversely, it could also increase the use of healthcare services as they increasingly seek reassurance of their health worries [28, 59].

Worrying about losing a loved one was also associated with health anxiety at T1, which has been previously linked to psychological distress [29]. Worrying about bills was associated with stress at T1. This finding is consistent with research demonstrating that economic shock, such as that caused by the COVID-19 pandemic, can predict stress [15, 60]. As stress can negatively

affect health and well-being [52, 61], financial worries are important to consider when managing stress in people with CKD, particularly for those at higher economic risk [62].

This study has several limitations. It is a cross-sectional study, and conclusions regarding associations are correlative rather than causative. Single items were also used to assess worry rather than a validated questionnaire which could affect the validity of the results. However, these items were taken from the WHO [36] and allowed us to look at worries in more detail, rather than as a collection or process. Worries about mental and physical health are also not likely to be entirely independent of the contents of the DASS-21 and SHAI. Although there was no evidence of multicollinearity for these relationships, conclusions regarding these associations should be interpreted cautiously. The population studied was largely homogenous as it comprised mostly White, older, and highly educated individuals, consequently, results are not generalisable to other populations who may experience worry and psychological distress differently [1, 2, 19].

The study has multiple strengths, including the use of validated questionnaires (the DASS-21 and SHAI) which have sound psychometric properties. This increases the reliability and validity of the results and allows for comparison with studies using these measures. This study was novel in evaluating the associations of particular worries with psychological distress in both an NDCKD and KTR population. Future research must assess the long-term impacts of the pandemic on the well-being of people with CKD, to improve health policy during post-pandemic recovery [63]. Additional factors that may mediate the relationship between worry and psychological distress such as social support should also be considered [15, 64, 65].

This study indicates that worries can have implications for psychological distress. HCPs should ensure that these worries are considered as part of forming a comprehensive understanding of the holistic care needs of people with CKD [66]. This can lead to signposting to appropriate care and support, as well as improving communication and relationships with patients [67].

## Conclusions

The findings of this study suggest worries such as mental health, physical health, ability to pay bills, and losing loved ones are associated with increased levels of psychological distress. Macro worries about economic recession and healthcare system overload were also prominent, which could negatively affect how people with CKD use healthcare services. These findings can assist HCPs to better understand which worries most affect people with CKD and how they are related to psychological distress, these worries should be considered in holistic care assessments to identify and work towards mitigating these concerns.

## Supporting information

**S1 Text. England (UK) COVID-19 lockdown and restrictions overview.**
(DOCX)

**S2 Text. Worry items revised from the WHO (2020) COVID-19 survey guidance.**
(DOCX)

**S1 Table. Mean rating of worry items (including "worry about becoming unemployed") for employed participants.**
(DOCX)

**S2 Table. Unadjusted multiple regression models.** Table A: Unadjusted model of association between worries and demographic factors at T1 and depression, anxiety, stress and health anxiety at T1; Table B: Unadjusted model of association between worries and demographic factors

at T2 and depression, anxiety, stress and health anxiety at T2.
(DOCX)

**S3 Table. Descriptive statistics of demographics for participants not included in the multiple regression analyses.**
(DOCX)

**S1 Data.**
(XLSX)

## Acknowledgments

This work is independent research funded by the Stoneygate Trust and the National Institute of Health and Care Research (NIHR) Applied Research Collaboration East Midlands and supported by the NIHR Leicester Biomedical Research Centre. The views expressed are those of the author(s) and not necessarily those of the Stoneygate Trust, NHS, NIHR ARC-EM and Leicester BRC. We would like to acknowledge the numerous research facilitators at each site who contributed to the collection of data in this study and other research staff and students who assisted with data collection, in particular Jared Palmer, Daniel Nixon, and Archontissa Kanavaki.

## Author Contributions

**Conceptualization:** Thomas J. Wilkinson, Alice C. Smith.

**Data curation:** Ella C. Ford, Gurneet K. Sohansoha, Roseanne E. Billany, Thomas J. Wilkinson, Courtney J. Lightfoot, Alice C. Smith.

**Formal analysis:** Ella C. Ford, Gurneet K. Sohansoha, Roseanne E. Billany, Thomas J. Wilkinson, Courtney J. Lightfoot.

**Funding acquisition:** Alice C. Smith.

**Investigation:** Ella C. Ford, Gurneet K. Sohansoha, Alice C. Smith.

**Methodology:** Ella C. Ford, Gurneet K. Sohansoha, Naeema A. Patel, Roseanne E. Billany, Thomas J. Wilkinson, Courtney J. Lightfoot, Alice C. Smith.

**Project administration:** Ella C. Ford, Gurneet K. Sohansoha, Roseanne E. Billany, Courtney J. Lightfoot, Alice C. Smith.

**Supervision:** Alice C. Smith.

**Writing – original draft:** Ella C. Ford.

**Writing – review & editing:** Ella C. Ford, Gurneet K. Sohansoha, Naeema A. Patel, Roseanne E. Billany, Thomas J. Wilkinson, Courtney J. Lightfoot, Alice C. Smith.

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
