## [Decision Letter · Decision Letter 0]

14 Aug 2024

The association of micro and macro worries with psychological distress in people living with chronic kidney disease during the COVID-19 pandemic

PONE-D-24-07870

Dear Prof Smith,

We’re pleased to inform you that your manuscript has been judged scientifically suitable for publication and will be formally accepted for publication once it meets all outstanding technical requirements.

Kind regards,

Collins Atta Poku

Academic Editor

PLOS ONE

2. We note that your Data Availability Statement is currently as follows: [All relevant data are within the manuscript and its Supporting Information files]

Reviewers' comments:

Reviewer's Responses to Questions

**Comments to the Author**

1. Is the manuscript technically sound, and do the data support the conclusions?

Reviewer #1: Yes

Reviewer #2: Yes

2. Has the statistical analysis been performed appropriately and rigorously? 

Reviewer #1: Yes

Reviewer #2: Yes

3. Have the authors made all data underlying the findings in their manuscript fully available?

Reviewer #1: Yes

Reviewer #2: Yes

4. Is the manuscript presented in an intelligible fashion and written in standard English?

Reviewer #1: Yes

Reviewer #2: Yes

5. Review Comments to the Author

Reviewer #1: The introduction is comprehensively written with appropriate references. The resources and techniques are thorough and reliable. The findings are given extra weight by the researchers' use of verified questionnaires. The findings are thorough and in-depth. The conversation was skillfully crafted, using wording that was suited for each situation. The results provide credence to the conclusion.

Reviewer #2: In my opinion, this manuscript describes a scientific research with an adequate design, original data that supports the conclusions. The study presented appropriately the conclusions based on the data presented.

6. PLOS authors have the option to publish the peer review history of their article (what does this mean?). If published, this will include your full peer review and any attached files.

Reviewer #1: **Yes: **Joseph Shahadu Issifu

Reviewer #2: No

---

## [Editor Report · Acceptance letter]

10 Oct 2024

PONE-D-24-07870 

PLOS ONE

Dear Dr. Smith, 

I'm pleased to inform you that your manuscript has been deemed suitable for publication in PLOS ONE. Congratulations! Your manuscript is now being handed over to our production team.

Kind regards, 

on behalf of

Dr. Collins Atta Poku 

Academic Editor

PLOS ONE